# Policy Poisoning
# in Batch Reinforcement Learning and Control

**Yuzhe Ma**
University of Wisconsin–Madison
 yzm234@cs.wisc.edu

**Xuezhou Zhang**
University of Wisconsin–Madison
zhangxz1123@cs.wisc.edu

**Wen Sun**
Microsoft Research New York
Sun.Wen@microsoft.com

**Xiaojin Zhu**
University of Wisconsin–Madison
jerryzhu@cs.wisc.edu

## Abstract

We study a security threat to batch reinforcement learning and control where the attacker aims to poison the learned policy. The victim is a reinforcement learner / controller which first estimates the dynamics and the rewards from a batch data set, and then solves for the optimal policy with respect to the estimates. The attacker can modify the data set slightly before learning happens, and wants to force the learner into learning a target policy chosen by the attacker. We present a unified framework for solving batch policy poisoning attacks, and instantiate the attack on two standard victims: tabular certainty equivalence learner in reinforcement learning and linear quadratic regulator in control. We show that both instantiation result in a convex optimization problem on which global optimality is guaranteed, and provide analysis on attack feasibility and attack cost. Experiments show the effectiveness of policy poisoning attacks.

## 1 Introduction

With the increasing adoption of machine learning, it is critical to study security threats to learning algorithms and design effective defense mechanisms against those threats. There has been significant work on adversarial attacks [2, 9]. We focus on the subarea of data poisoning attacks where the adversary manipulates the training data so that the learner learns a wrong model. Prior work on data poisoning targeted victims in supervised learning [17, 13, 19, 22] and multi-armed bandits [11, 16, 15]. We take a step further and study data poisoning attacks on reinforcement learning (RL). Given RL's prominent applications in robotics, games and so on, an intentionally and adversarially planted bad policy could be devastating.

While there has been some related work in test-time attack on RL, reward shaping, and teaching inverse reinforcement learning (IRL), little is understood on how to training-set poison a reinforcement learner. We take the first step and focus on *batch* reinforcement learner and controller as the victims. These victims learn their policy from a batch training set. We assume that the attacker can modify the rewards in the training set, which we show is sufficient for policy poisoning. The attacker's goal is to force the victim to learn a particular target policy (hence the name policy poisoning), while minimizing the reward modifications. Our main contribution is to characterize batch policy poisoning with a unified optimization framework, and to study two instances against tabular certainty-equivalence (TCE) victim and linear quadratic regulator (LQR) victim, respectively.

## 2   Related Work

Of particular interest is the work on *test-time attacks* against RL [10]. Unlike policy poisoning, there the RL agent carries out an already-learned and fixed policy $\pi$ to e.g. play the Pong Game. The attacker perturbs pixels in a game board image, which is part of the state $s$. This essentially changes the RL agent's perceived state into some $s'$. The RL agent then chooses the action $a' := \pi(s')$ (e.g. move down) which may differ from $a := \pi(s)$ (e.g. move up). The attacker's goal is to force some specific $a'$ on the RL agent. Note $\pi$ itself stays the same through the attack. In contrast, ours is a data-poisoning attack which happens at training time and aims to change $\pi$.

Data-poisoning attacks were previously limited to supervised learning victims, either in batch mode [3, 21, 14, 17] or online mode [19, 22]. Recently data-poisoning attacks have been extended to multi-armed bandit victims [11, 16, 15], but not yet to RL victims.

There are two related but distinct concepts in RL research. One concept is reward shaping [18, 1, 7, 20] which also modifies rewards to affect an RL agent. However, the goal of reward shaping is fundamentally different from ours. Reward shaping aims to speed up convergence to the *same* optimal policy as without shaping. Note the differences in both the target (same vs. different policies) and the optimality measure (speed to converge vs. magnitude of reward change).

The other concept is teaching IRL [5, 4, 12]. Teaching and attacking are mathematically equivalent. However, the main difference to our work is the victim. They require an IRL agent, which is a specialized algorithm that estimates a reward function from demonstrations of (state, action) trajectories alone (i.e. no reward given). In contrast, our attacks target more prevalent RL agents and are thus potentially more applicable. Due to the difference in the input to IRL vs. RL victims, our attack framework is completely different.

## 3   Preliminaries

A Markov Decision Process (MDP) is defined as a tuple $(\mathcal{S}, \mathcal{A}, P, R, \gamma)$, where $\mathcal{S}$ is the state space, $\mathcal{A}$ is the action space, $P : \mathcal{S} \times \mathcal{A} \to \Delta_{\mathcal{S}}$ is the transition kernel where $\Delta_{\mathcal{S}}$ denotes the space of probability distributions on $\mathcal{S}$, $R : \mathcal{S} \times \mathcal{A} \to \mathbb{R}$ is the reward function, and $\gamma \in [0, 1)$ is a discounting factor. We define a policy $\pi : \mathcal{S} \to \mathcal{A}$ as a function that maps a state to an action. We denote the $Q$ function of a policy $\pi$ as $Q^{\pi}(s, a) = \mathbb{E}[\sum_{\tau=0}^{\infty} \gamma^{\tau} R(s_{\tau}, a_{\tau}) \mid s_0 = s, a_0 = a, \pi]$, where the expectation is over the randomness in both transitions and rewards. The $Q$ function that corresponds to the optimal policy can be characterized by the following Bellman optimality equation:

$$Q^*(s, a) = R(s, a) + \gamma \sum_{s' \in \mathcal{S}} P(s'|s, a) \max_{a' \in \mathcal{A}} Q^*(s', a'), \tag{1}$$

and the optimal policy is defined as $\pi^*(s) \in \arg\max_{a \in \mathcal{A}} Q^*(s, a)$.

We focus on RL victims who perform batch reinforcement learning. A training item is a tuple $(s, a, r, s') \in \mathcal{S} \times \mathcal{A} \times \mathbb{R} \times \mathcal{S}$, where $s$ is the current state, $a$ is the action taken, $r$ is the received reward, and $s'$ is the next state. A training set is a batch of $T$ training items denoted by $D = (s_t, a_t, r_t, s'_t)_{t=0:T-1}$. Given training set $D$, a model-based learner performs learning in two steps:

**Step 1**. The learner estimates an MDP $\hat{M} = (\mathcal{S}, \mathcal{A}, \hat{P}, \hat{R}, \gamma)$ from $D$. In particular, we assume the learner uses maximum likelihood estimate for the transition kernel $\hat{P} : \mathcal{S} \times \mathcal{A} \mapsto \Delta_{\mathcal{S}}$

$$\hat{P} \quad \in \quad \arg\max_{P} \sum_{t=0}^{T-1} \log P(s'_t|s_t, a_t), \tag{2}$$

and least-squares estimate for the reward function $\hat{R} : \mathcal{S} \times \mathcal{A} \mapsto \mathbb{R}$

$$\hat{R} \quad = \quad \arg\min_{R} \sum_{t=0}^{T-1} (r_t - R(s_t, a_t))^2. \tag{3}$$

Note that we do not require (2) to have a unique maximizer $\hat{P}$. When multiple maximizers exist, we assume the learner arbitrarily picks one of them as the estimate. We assume the minimizer $\hat{R}$ is always unique. We will discuss the conditions to guarantee the uniqueness of $\hat{R}$ for two learners later.

**Step 2**. The learner finds the optimal policy $\hat{\pi}$ that maximizes the expected discounted cumulative reward on the estimated environment $\hat{M}$, i.e.,

$$\hat{\pi} \in \arg\max_{\pi:\mathcal{S}\mapsto\mathcal{A}} \mathbb{E}_{\hat{P}} \sum_{\tau=0}^{\infty} \gamma^{\tau} \hat{R}(s_{\tau}, \pi(s_{\tau})), \tag{4}$$

where $s_0$ is a specified or random initial state. Note that there could be multiple optimal policies, thus we use $\in$ in (4). Later we will specialize (4) to two specific victim learners: the tabular certainty equivalence learner (TCE) and the certainty-equivalent linear quadratic regulator (LQR).

# 4 Policy Poisoning

We study policy poisoning attacks on model-based batch RL learners. Our threat model is as follows:

**Knowledge of the attacker.** The attacker has access to the original training set $D^0 = (s_t, a_t, r_t^0, s_t')_{t=0:T-1}$. The attacker knows the model-based RL learner's algorithm. Importantly, the attacker knows how the learner estimates the environment, i.e., (2) and (3). In the case (2) has multiple maximizers, we assume the attacker knows exactly the $\hat{P}$ that the learner picks.

**Available actions of the attacker.** The attacker is allowed to arbitrarily modify the rewards $\mathbf{r}^0 = (r_0^0, ..., r_{T-1}^0)$ in $D^0$ into $\mathbf{r} = (r_0, ..., r_{T-1})$. As we show later, changing $r$'s but not $s, a, s'$ is sufficient for policy poisoning.

**Attacker's goals.** The attacker has a pre-specified target policy $\pi^{\dagger}$. The attack goals are to (1) force the learner to learn $\pi^{\dagger}$, (2) minimize attack cost $\|\mathbf{r} - \mathbf{r}^0\|_{\alpha}$ under an $\alpha$-norm chosen by the attacker.

Given the threat model, we can formulate policy poisoning as a bi-level optimization problem[1]:

$$\min_{\mathbf{r}, \hat{R}} \quad \|\mathbf{r} - \mathbf{r}^0\|_{\alpha} \tag{5}$$

$$\text{s.t.} \quad \hat{R} = \arg\min_{R} \sum_{t=0}^{T-1} (r_t - R(s_t, a_t))^2 \tag{6}$$

$$\{\pi^{\dagger}\} = \arg\max_{\pi:\mathcal{S}\mapsto\mathcal{A}} \mathbb{E}_{\hat{P}} \sum_{\tau=0}^{\infty} \gamma^{\tau} \hat{R}(s_{\tau}, \pi(s_{\tau})). \tag{7}$$

The $\hat{P}$ in (7) does not involve $\mathbf{r}$ and is precomputed from $D^0$. The singleton set $\{\pi^{\dagger}\}$ on the LHS of (7) ensures that the target policy is learned uniquely, i.e., there are no other optimal policies tied with $\pi^{\dagger}$. Next, we instantiate this attack formulation to two representative model-based RL victims.

## 4.1 Poisoning a Tabular Certainty Equivalence (TCE) Victim

In tabular certainty equivalence (TCE), the environment is a Markov Decision Process (MDP) with finite state and action space. Given original data $D^0 = (s_t, a_t, r_t^0, s_t')_{0:T-1}$, let $T_{s,a} = \{t \mid s_t = s, a_t = a\}$, the time indexes of all training items for which action $a$ is taken at state $s$. We assume $T_{s,a} \geq 1, \forall s, a$, i.e., each state-action pair appears at least once in $D^0$. This condition is needed to ensure that the learner's estimate $\hat{P}$ and $\hat{R}$ exist. Remember that we require (3) to have a unique solution. For the TCE learner, $\hat{R}$ is unique as long as it exists. Therefore, $T_{s,a} \geq 1, \forall s, a$ is sufficient to guarantee a unique solution to (3). Let the poisoned data be $D = (s_t, a_t, r_t, s_t')_{0:T-1}$. Instantiating model estimation (2), (3) for TCE, we have

$$\hat{P}(s' \mid s, a) = \frac{1}{|T_{s,a}|} \sum_{t \in T_{s,a}} \mathbb{1}\left[s_t' = s'\right], \tag{8}$$

where $\mathbb{1}[]$ is the indicator function, and

$$\hat{R}(s, a) = \frac{1}{|T_{s,a}|} \sum_{t \in T_{s,a}} r_t. \tag{9}$$

The TCE learner uses $\hat{P}, \hat{R}$ to form an estimated MDP $\hat{M}$, then solves for the optimal policy $\hat{\pi}$ with respect to $\hat{M}$ using the Bellman equation (1). The attack goal (7) can be naively characterized by

$$Q(s, \pi^\dagger(s)) > Q(s, a), \forall s \in \mathcal{S}, \forall a \neq \pi^\dagger(s). \tag{10}$$

However, due to the strict inequality, (10) induces an open set in the $Q$ space, on which the minimum of (5) may not be well-defined. Instead, we require a stronger attack goal which leads to a closed subset in the $Q$ space. This is defined as the following $\varepsilon$-robust target $Q$ polytope.

**Definition 1.** *($\varepsilon$-robust target $Q$ polytope) The set of $\varepsilon$-robust $Q$ functions induced by a target policy $\pi^\dagger$ is the polytope*

$$\mathcal{Q}_\varepsilon(\pi^\dagger) = \{Q : Q(s, \pi^\dagger(s)) \geq Q(s, a) + \varepsilon, \forall s \in \mathcal{S}, \forall a \neq \pi^\dagger(s)\} \tag{11}$$

*for a fixed $\varepsilon > 0$.*

The margin parameter $\varepsilon$ ensures that $\pi^\dagger$ is the unique optimal policy for any $Q$ in the polytope. We now have a solvable attack problem, where the attacker wants to force the victim's $Q$ function into the $\varepsilon$-robust target $Q$ polytope $\mathcal{Q}_\varepsilon(\pi^\dagger)$:

$$\min_{\mathbf{r} \in \mathbb{R}^T, \hat{R}, Q \in \mathbb{R}^{|\mathcal{S}| \times |\mathcal{A}|}} \quad \|\mathbf{r} - \mathbf{r}^0\|_\alpha \tag{12}$$

$$\text{s.t.} \quad \hat{R}(s, a) = \frac{1}{|T_{s,a}|} \sum_{t \in T_{s,a}} r_t \tag{13}$$

$$Q(s, a) = \hat{R}(s, a) + \gamma \sum_{s'} \hat{P}\left(s'|s, a\right) Q\left(s', \pi^\dagger(s')\right), \forall s, \forall a \tag{14}$$

$$Q(s, \pi^\dagger(s)) \geq Q(s, a) + \varepsilon, \forall s \in \mathcal{S}, \forall a \neq \pi^\dagger(s). \tag{15}$$

The constraint (14) enforces Bellman optimality on the value function $Q$, in which $\max_{a' \in \mathcal{A}} Q(s', a')$ is replaced by $Q\left(s', \pi^\dagger(s')\right)$, since the target policy is guaranteed to be optimal by (15). Note that problem (12)-(15) is a convex program with linear constraints given that $\alpha \geq 1$, thus could be solved to global optimality. However, we point out that (12)-(15) is a more stringent formulation than (5)-(7) due to the additional margin parameter $\varepsilon$ we introduced. The feasible set of (12)-(15) is a subset of (5)-(7). Therefore, the optimal solution to (12)-(15) could in general be a sub-optimal solution to (5)-(7) with potentially larger objective value. We now study a few theoretical properties of policy poisoning on TCE. All proofs are in the appendix. First of all, the attack is always feasible.

**Proposition 1.** *The attack problem* (12)-(15) *is always feasible for any target policy $\pi^\dagger$.*

Proposition 1 states that for any target policy $\pi^\dagger$, there exists a perturbation on the rewards that teaches the learner that policy. Therefore, the attacker changing $r$'s but not $s, a, s'$ is already sufficient for policy poisoning.

We next bound the attack cost. Let the MDP estimated on the clean data be $\hat{M}^0 = (\mathcal{S}, \mathcal{A}, \hat{P}, \hat{R}^0, \gamma)$. Let $Q^0$ be the $Q$ function that satisfies the Bellman optimality equation on $\hat{M}^0$. Define $\Delta(\varepsilon) = \max_{s \in \mathcal{S}}[\max_{a \neq \pi^\dagger(s)} Q^0(s, a) - Q^0(s, \pi^\dagger(s)) + \varepsilon]_+$, where $[]_+$ takes the maximum over 0. Intuitively, $\Delta(\varepsilon)$ measures how suboptimal the target policy $\pi^\dagger$ is compared to the clean optimal policy $\pi^0$ learned on $\hat{M}^0$, up to a margin parameter $\varepsilon$.

**Theorem 2.** *Assume $\alpha \geq 1$ in* (12)*. Let $\mathbf{r}^*$, $\hat{R}^*$ and $Q^*$ be an optimal solution to* (12)-(15)*, then*

$$\frac{1}{2}(1 - \gamma)\Delta(\varepsilon) \left(\min_{s,a} |T_{s,a}|\right)^{\frac{1}{\alpha}} \leq \|\mathbf{r}^* - \mathbf{r}^0\|_\alpha \leq \frac{1}{2}(1 + \gamma)\Delta(\varepsilon) T^{\frac{1}{\alpha}}. \tag{16}$$

**Corollary 3.** *If $\alpha = 1$, then the optimal attack cost is $O(\Delta(\varepsilon)T)$. If $\alpha = 2$, then the optimal attack cost is $O(\Delta(\varepsilon)\sqrt{T})$. If $\alpha = \infty$, then the optimal attack cost is $O(\Delta(\varepsilon))$.*

Note that both the upper and lower bounds on the attack cost are linear with respect to $\Delta(\varepsilon)$, which can be estimated directly from the clean training set $D^0$. This allows the attacker to easily estimate its attack cost before actually solving the attack problem.

## 4.2 Poisoning a Linear Quadratic Regulator (LQR) Victim

As the second example, we study an LQR victim that performs system identification from a batch training set [6]. Let the linear dynamical system be

$$s_{t+1} = As_t + Ba_t + w_t, \forall t \geq 0, \tag{17}$$

where $A \in \mathbb{R}^{n \times n}, B \in \mathbb{R}^{n \times m}$, $s_t \in \mathbb{R}^n$ is the state, $a_t \in \mathbb{R}^m$ is the control signal, and $w_t \sim \mathcal{N}(\mathbf{0}, \sigma^2 I)$ is a Gaussian noise. When the agent takes action $a$ at state $s$, it suffers a quadratic loss of the general form

$$L(s, a) = \frac{1}{2} s^\top Q s + q^\top s + a^\top R a + c \tag{18}$$

for some $Q \succeq 0$, $R \succ 0$, $q \in \mathbb{R}^n$ and $c \in \mathbb{R}$. Here we have redefined the symbols $Q$ and $R$ in order to conform with the notation convention in LQR: now we use $Q$ for the quadratic loss matrix associated with state, not the action-value function; we use $R$ for the quadratic loss matrix associated with action, not the reward function. The previous reward function $R(s, a)$ in general MDP (section 3) is now equivalent to the negative loss $-L(s, a)$. This form of loss captures various LQR control problems. Note that the above linear dynamical system can be viewed as an MDP with transition kernel $P(s' \mid s, a) = \mathcal{N}(As + Ba, \sigma^2 I)$ and reward function $-L(s, a)$. The environment is thus characterized by matrices $A$, $B$ (for transition kernel) and $Q$, $R$, $q$, $c$ (for reward function), which are all unknown to the learner.

We assume the clean training data $D^0 = (s_t, a_t, r_t^0, s_{t+1})_{0:T-1}$ was generated by running the linear system for multiple episodes following some random policy [6]. Let the poisoned data be $D = (s_t, a_t, r_t, s_{t+1})_{0:T-1}$. Instantiating model estimation (2), (3), the learner performs system identification on the poisoned data:

$$(\hat{A}, \hat{B}) \in \underset{(A,B)}{\arg\min} \frac{1}{2} \sum_{t=0}^{T-1} \|As_t + Ba_t - s_{t+1}\|_2^2 \tag{19}$$

$$(\hat{Q}, \hat{R}, \hat{q}, \hat{c}) = \underset{(Q \succeq 0, R \succeq \varepsilon I, q, c)}{\arg\min} \frac{1}{2} \sum_{t=0}^{T-1} \left\| \frac{1}{2} s_t^\top Q s_t + q^\top s_t + a_t^\top R a_t + c + r_t \right\|_2^2. \tag{20}$$

Note that in (20), the learner uses a stronger constraint $R \succeq \varepsilon I$ than the original constraint $R \succ 0$, which guarantees that the minimizer can be achieved. The conditions to further guarantee (20) having a unique solution depend on the property of certain matrices formed by the clean training set $D^0$, which we defer to appendix D.

The learner then computes the optimal control policy with respect to $\hat{A}$, $\hat{B}$, $\hat{Q}$, $\hat{R}$, $\hat{q}$ and $\hat{c}$. We assume the learner solves a discounted version of LQR control

$$\max_{\pi:\mathcal{S} \mapsto \mathcal{A}} \quad -\mathbb{E}\left[ \sum_{\tau=0}^{\infty} \gamma^\tau (\frac{1}{2} s_\tau^\top \hat{Q} s_\tau + \hat{q}^\top s_\tau + \pi(s_\tau)^\top \hat{R} \pi(s_\tau) + \hat{c}) \right] \tag{21}$$

$$\text{s.t.} \quad s_{\tau+1} = \hat{A}s_\tau + \hat{B}\pi(s_\tau) + w_\tau, \forall \tau \geq 0. \tag{22}$$

where the expectation is over $w_\tau$. It is known that the control problem has a closed-form solution $\hat{a}_\tau = \hat{\pi}(s_\tau) = Ks_\tau + k$, where

$$K = -\gamma \left( \hat{R} + \gamma \hat{B}^\top X \hat{B} \right)^{-1} \hat{B}^\top X \hat{A}, \quad k = -\gamma(\hat{R} + \gamma \hat{B}^\top X \hat{B})^{-1} \hat{B}^\top x. \tag{23}$$

Here $X \succeq 0$ is the unique solution of the Algebraic Riccati Equation,

$$X = \gamma \hat{A}^\top X \hat{A} - \gamma^2 \hat{A}^\top X \hat{B} \left( \hat{R} + \gamma \hat{B}^\top X \hat{B} \right)^{-1} \hat{B}^\top X \hat{A} + \hat{Q}, \tag{24}$$

and $x$ is a vector that satisfies

$$x = \hat{q} + \gamma(\hat{A} + \hat{B}K)^\top x. \tag{25}$$

The attacker aims to force the victim into taking target action $\pi^\dagger(s), \forall s \in \mathbb{R}^n$. Note that in LQR, the attacker cannot arbitrarily choose $\pi^\dagger$, as the optimal control policy $K$ and $k$ enforce a linear structural constraint between $\pi^\dagger(s)$ and $s$. One can easily see that the target action must obey $\pi^\dagger(s) = K^\dagger s + k^\dagger$

for some $(K^\dagger, k^\dagger)$ in order to achieve successful attack. Therefore we must assume instead that the attacker has a target policy specified by a pair $(K^\dagger, k^\dagger)$. However, an arbitrarily linear policy may still not be feasible. A target policy $(K^\dagger, k^\dagger)$ is feasible if and only if it is produced by solving some Riccati equation, namely, it must lie in the following set:

$$\{(K, k) : \exists Q \succeq 0, R \succeq \varepsilon I, q \in \mathbb{R}^n, c \in \mathbb{R}, \text{ such that (23), (24), and (25) are satisfied}\}. \quad (26)$$

Therefore, to guarantee feasibility, we assume the attacker always picks the target policy $(K^\dagger, k^\dagger)$ by solving an LQR problem with some attacker-defined loss function. We can now pose the policy poisoning attack problem:

$$\min_{\mathbf{r}, \hat{Q}, \hat{R}, \hat{q}, \hat{c}, X, x} \quad \|\mathbf{r} - \mathbf{r}^0\|_\alpha \quad (27)$$

$$\text{s.t.} \quad -\gamma \left( \hat{R} + \gamma \hat{B}^\top X \hat{B} \right)^{-1} \hat{B}^\top X \hat{A} = K^\dagger \quad (28)$$

$$-\gamma \left( \hat{R} + \gamma \hat{B}^\top X \hat{B} \right)^{-1} \hat{B}^\top x = k^\dagger \quad (29)$$

$$X = \gamma \hat{A}^\top X \hat{A} - \gamma^2 \hat{A}^\top X \hat{B} \left( \hat{R} + \gamma \hat{B}^\top X \hat{B} \right)^{-1} \hat{B}^\top X \hat{A} + \hat{Q} \quad (30)$$

$$x = \hat{q} + \gamma (\hat{A} + \hat{B} K^\dagger)^\top x \quad (31)$$

$$(\hat{Q}, \hat{R}, \hat{q}, \hat{c}) = \arg\min_{(Q \succeq 0, R \succeq \varepsilon I, q, c)} \sum_{t=0}^{T-1} \left\| \frac{1}{2} s_t^\top Q s_t + q^\top s_t + a_t^\top R a_t + c + r_t \right\|_2^2 \quad (32)$$

$$X \succeq 0. \quad (33)$$

Note that the estimated transition matrices $\hat{A}$, $\hat{B}$ are not optimization variables because the attacker can only modify the rewards, which will not change the learner's estimate on $\hat{A}$ and $\hat{B}$. The attack optimization (27)-(33) is hard to solve due to the constraint (32) itself being a semi-definite program (SDP). To overcome the difficulty, we pull all the positive semi-definite constraints out of the lower-level optimization. This leads to a more stringent surrogate attack optimization (see appendix C). Solving the surrogate attack problem, whose feasible region is a subset of the original problem, in general gives a suboptimal solution to (27)-(33). But it comes with one advantage: convexity.

## 5 Experiments

Throughout the experiments, we use CVXPY [8] to implement the optimization. All code can be found in https://github.com/myzwisc/PPRL_NeurIPS19.

### 5.1 Policy Poisoning Attack on TCE Victim

**Experiment 1.** We consider a simple MDP with two states $A$, $B$ and two actions: *stay* in the same state or *move* to the other state, shown in figure 1a. The discounting factor is $\gamma = 0.9$. The MDP's $Q$ values are shown in table 1b. Note that the optimal policy will always pick action *stay*. The clean training data $D^0$ reflects this underlying MDP, and consists of 4 tuples:

$$(A, stay, 1, A) \quad (A, move, 0, B) \quad (B, stay, 1, B) \quad (B, move, 0, A)$$

Let the attacker's target policy be $\pi^\dagger(s) = move$, for any state $s$. The attacker sets $\varepsilon = 1$ and uses $\alpha = 2$, i.e. $\|\mathbf{r} - \mathbf{r}^0\|_2$ as the attack cost. Solving the policy poisoning attack optimization problem (12)-(15) produces the poisoned data:

$$(A, stay, 0, A) \quad (A, move, 1, B) \quad (B, stay, 0, B) \quad (B, move, 1, A)$$

with attack cost $\|\mathbf{r} - \mathbf{r}^0\|_2 = 2$. The resulting poisoned $Q$ values are shown in table 1c. To verify this attack, we run TCE learner on both clean data and poisoned data. Specifically, we estimate the transition kernel and the reward function as in (8) and (9) on each data set, and then run value iteration until the $Q$ values converge. In Figure 1d, we show the trajectory of $Q$ values for state $A$, where the $x$ and $y$ axes denote $Q(A, stay)$ and $Q(A, move)$ respectively. All trajectories start at $(0, 0)$. The dots on the trajectory correspond to each step of value iteration, while the star denotes the converged $Q$ values. The diagonal dashed line is the (zero margin) policy boundary, while the gray

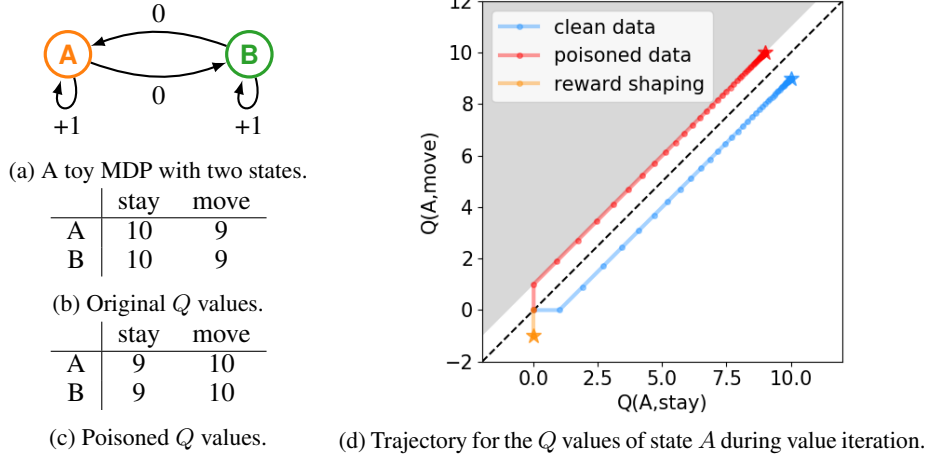

(a) A toy MDP with two states.

|   | stay | move |
|---|------|------|
| A | 10   | 9    |
| B | 10   | 9    |

(b) Original $Q$ values.

|   | stay | move |
|---|------|------|
| A | 9    | 10   |
| B | 9    | 10   |

(c) Poisoned $Q$ values.

(d) Trajectory for the $Q$ values of state $A$ during value iteration.

Figure 1: Poisoning TCE in a two-state MDP.

region is the $\varepsilon$-robust target $Q$ polytope with an offset $\varepsilon = 1$ to the policy boundary. The trajectory of clean data converges to a point below the policy boundary, where the action $stay$ is optimal. With the poisoned data, the trajectory of $Q$ values converge to a point exactly on the boundary of the $\varepsilon$-robust target $Q$ polytope, where the action $move$ becomes optimal. This validates our attack.

We also compare our attack with reward shaping [18]. We let the potential function $\phi(s)$ be the optimal value function $V(s)$ for all $s$ to shape the clean dataset. The dataset after shaping is

$$(A, stay, 0, A) \quad (A, move, -1, B) \quad (B, stay, 0, B) \quad (B, move, -1, A)$$

In Figure 1d, we show the trajectory of $Q$ values after reward shaping. Note that same as on clean dataset, the trajectory after shaping converges to a point also below the policy boundary. This means reward shaping can not make the learner learn a different policy from the original optimal policy. Also note that after reward shaping, value iteration converges much faster (in only one iteration), which matches the benefits of reward shaping shown in [18]. More importantly, this illustrates the difference between our attack and reward shaping.

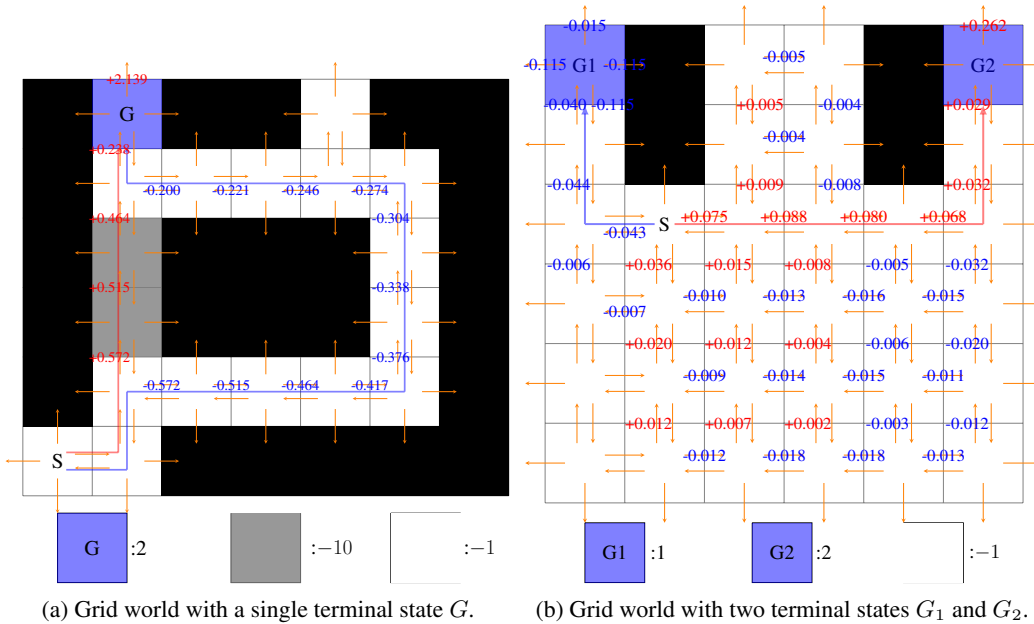

(a) Grid world with a single terminal state $G$.

(b) Grid world with two terminal states $G_1$ and $G_2$.

Figure 2: Poisoning TCE in grid-world tasks.

**Experiment 2.** As another example, we consider the grid world tasks in [5]. In particular, we focus on two tasks shown in figure 2a and 2b. In figure 2a, the agent starts from S and aims to arrive at the terminal cell G. The black regions are walls, thus the agent can only choose to go through the white or gray regions. The agent can take four actions in every state: go left, right, up or down, and stays if the action takes it into the wall. Reaching a gray, white, or the terminal state results in rewards $-10$, $-1$, 2, respectively. After the agent arrives at the terminal state G, it will stay there forever and always receive reward 0 regardless of the following actions. The original optimal policy is to follow the blue trajectory. The attacker's goal is to force the agent to follow the red trajectory. Correspondingly, we set the target actions for those states on the red trajectory as along the trajectory. We set the target actions for the remaining states to be the same as the original optimal policy learned on clean data.

The clean training data contains a single item for every state-action pair. We run the attack with $\varepsilon = 0.1$ and $\alpha = 2$. Our attack is successful: with the poisoned data, TCE generates a policy that produces the red trajectory in Figure 2a, which is the desired behavior. The attack cost is $\|\mathbf{r} - \mathbf{r}^0\|_2 \approx 2.64$, which is small compared to $\|\mathbf{r}^0\|_2 = 21.61$. In Figure 2a, we show the poisoning on rewards. Each state-action pair is denoted by an orange arrow. The value tagged to each arrow is the modification to that reward, where red value means the reward is increased and blue means decreased. An arrow without any tagged value means the corresponding reward is not changed by attack. Note that rewards along the red trajectory are increased, while those along the blue trajectory are reduced, resulting in the red trajectory being preferred by the agent. Furthermore, rewards closer to the starting state S suffer larger poisoning since they contribute more to the $Q$ values. For the large attack +2.139 happening at terminal state, we provide an explanation in appendix E.

**Experiment 3.** In Figure 2b there are two terminal states G1 and G2 with reward 1 and 2, respectively. The agent starts from S. Although G2 is more profitable, the path is longer and each step has a $-1$ reward. Therefore, the original optimal policy is the blue trajectory to G1. The attacker's target policy is to force the agent along the red trajectory to G2. We set the target actions for states as in experiment 2. The clean training data contains a single item for every state-action pair. We run our attack with $\varepsilon = 0.1$ and $\alpha = 2$. Again, after the attack, TCE on the poisoned dataset produces the red trajectory in figure 2b, with attack cost $\|\mathbf{r} - \mathbf{r}^0\|_2 \approx 0.38$, compared to $\|\mathbf{r}^0\|_2 = 11.09$. The reward poisoning follows a similar pattern to experiment 2.

## 5.2 Policy Poisoning Attack on LQR Victim

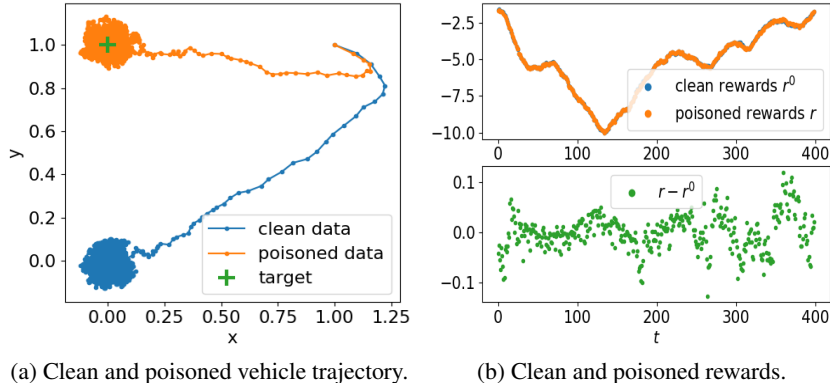

(a) Clean and poisoned vehicle trajectory.　　　(b) Clean and poisoned rewards.

Figure 3: Poisoning a vehicle running LQR in 4D state space.

**Experiment 4.** We now demonstrate our attack on LQR. We consider a linear dynamical system that approximately models a vehicle. The state of the vehicle consists of its 2D position and 2D velocity: $s_t = (x_t, y_t, v_t^x, v_t^y) \in \mathbb{R}^4$. The control signal at time $t$ is the force $a_t \in \mathbb{R}^2$ which will be applied on the vehicle for $h$ seconds. We assume there is a friction parameter $\eta$ such that the friction force is $-\eta v_t$. Let $m$ be the mass of the vehicle. Given small enough $h$, the transition matrices can be approximated by (17) where

$$A = \begin{bmatrix} 1 & 0 & h & 0 \\ 0 & 1 & 0 & h \\ 0 & 0 & 1 - h\eta/m & 0 \\ 0 & 0 & 0 & 1 - h\eta/m \end{bmatrix}, B = \begin{bmatrix} 0 & 0 \\ 0 & 0 \\ h/m & 0 \\ 0 & h/m \end{bmatrix}. \tag{34}$$

In this example, we let $h = 0.1$, $m = 1$, $\eta = 0.5$, and $w_t \sim \mathcal{N}(0, \sigma^2 I)$ with $\sigma = 0.01$. The vehicle starts from initial position $(1, 1)$ with velocity $(1, -0.5)$, i.e., $s_0 = (1, 1, 1, -0.5)$. The true loss function is $L(s, a) = \frac{1}{2} s^\top Q s + a^\top R a$ with $Q = I$ and $R = 0.1I$ (i.e., $Q = I, R = 0.1I, q = 0, c = 0$ in (18)). Throughout the experiment, we let $\gamma = 0.9$ for solving the optimal control policy in (21). With the true dynamics and loss function, the computed optimal control policy is

$$K^* = \begin{bmatrix} -1.32 & 0 & -2.39 & 0 \\ 0 & -1.32 & 0 & -2.39 \end{bmatrix}, k^* = [\ 0 \quad 0\ ], \tag{35}$$

which will drive the vehicle to the origin.

The batch LQR learner estimates the dynamics and the loss function from a batch training data. To produce the training data, we let the vehicle start from state $s_0$ and simulate its trajectory with a random control policy. Specifically, in each time step, we uniformly sample a control signal $a_t$ in a unit sphere. The vehicle then takes action $a_t$ to transit from current state $s_t$ to the next state $s_{t+1}$, and receives a reward $r_t = -L(s_t, a_t)$. This gives us one training item $(s_t, a_t, r_t, s_{t+1})$. We simulate a total of 400 time steps to obtain a batch data that contains 400 items, on which the learner estimates the dynamics and the loss function. With the learner's estimate, the computed clean optimal policy is

$$\hat{K}^0 = \begin{bmatrix} -1.31 & 1.00\mathrm{e}{-2} & -2.41 & 2.03\mathrm{e}{-3} \\ -1.97\mathrm{e}{-2} & -1.35 & -1.14\mathrm{e}{-2} & -2.42 \end{bmatrix}, \hat{k}^0 = [\ -4.88\mathrm{e}{-5} \quad 4.95\mathrm{e}{-6}\ ]. \tag{36}$$

The clean optimal policy differs slightly from the true optimal policy due to the inaccuracy of the learner's estimate. The attacker has a target policy $(K^\dagger, k^\dagger)$ that can drive the vehicle close to its target destination $(x^\dagger, y^\dagger) = (0, 1)$ with terminal velocity $(0, 0)$, which can be represented as a target state $s^\dagger = (0, 1, 0, 0)$. To ensure feasibility, we assume that the attacker starts with the loss function $\frac{1}{2}(s - s^\dagger)^\top Q(s - s^\dagger) + a^\top R a$ where $Q = I, R = 0.1I$. Due to the offset this corresponds to setting $Q = I, R = 0.1I, q = -s^\dagger, c = \frac{1}{2}s^{\dagger\top} Q s^\dagger = 0.5$ in (18). The attacker then solves the Riccati equation with its own loss function and the learner's estimates $\hat{A}$ and $\hat{B}$ to arrive at the target policy

$$K^\dagger = \begin{bmatrix} -1.31 & 9.99\mathrm{e}{-3} & -2.41 & 2.02\mathrm{e}{-3} \\ -1.97\mathrm{e}{-2} & -1.35 & -1.14\mathrm{e}{-2} & -2.42 \end{bmatrix}, k^\dagger = [\ -0.01 \quad 1.35\ ]. \tag{37}$$

We run our attack (27)-(33) with $\alpha = 2$ and $\varepsilon = 0.01$ in (32). Figure 3 shows the result of our attack. In Figure 3a, we plot the trajectory of the vehicle with policy learned on clean data and poisoned data respectively. Our attack successfully forces LQR into a policy that drives the vehicle close to the target destination. The wiggle on the trajectory is due to the noise $w_t$ of the dynamical system. On the poisoned data, the LQR victim learns the policy

$$\hat{K} = \begin{bmatrix} -1.31 & 9.99\mathrm{e}{-3} & -2.41 & 2.02\mathrm{e}{-3} \\ -1.97\mathrm{e}{-2} & -1.35 & -1.14\mathrm{e}{-2} & -2.42 \end{bmatrix}, \hat{k} = [\ -0.01 \quad 1.35\ ], \tag{38}$$

which matches exactly the target policy $K^\dagger, k^\dagger$. In Figure 3b, we show the poisoning on rewards. Our attack leads to very small modification on each reward, thus the attack is efficient. The total attack cost over all 400 items is only $\|\mathbf{r} - \mathbf{r}^0\|_2 = 0.73$, which is tiny small compared to $\|\mathbf{r}^0\|_2 = 112.94$. The results here demonstrate that our attack can dramatically change the behavior of LQR by only slightly modifying the rewards in the dataset.

Finally, for both attacks on TCE and LQR, we note that by setting the attack cost norm $\alpha = 1$ in (5), the attacker is able to obtain a *sparse* attack, meaning that only a small fraction of the batch data needs to be poisoned. Such sparse attacks have profound implications in adversarial machine learning as they can be easier to carry out and harder to detect. We show detailed results in appendix E.

# 6 Conclusion

We presented a policy poisoning framework against batch reinforcement learning and control. We showed the attack problem can be formulated as convex optimization. We provided theoretical analysis on attack feasibility and cost. Experiments show the attack can force the learner into an attacker-chosen target policy while incurring only a small attack cost.

**Acknowledgments.** This work is supported in part by NSF 1545481, 1561512, 1623605, 1704117, 1836978 and the MADLab AF Center of Excellence FA9550-18-1-0166.

## Footnotes

[1] As we will show, the constraint (7) could lead to an open feasible set (e.g., in (10)) for the attack optimization (5)-(7), on which the minimum of the objective function (5) may not be well-defined. In the case (7) induces an open set, we will consider instead a closed subset of it, and optimize over the subset. How to construct the closed subset will be made clear for concrete learners later.

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
