[Supplementary Material]

# Supplementary Material

## A  Proof of Proposition 1

The proof of feasibility relies on the following result, which states that there is a bijection mapping between reward space and value function space.

**Proposition 4.** *Given an MDP with transition probability function $P$ and discounting factor $\gamma \in [0, 1)$, let $\mathcal{R} = \{R : \mathcal{S} \times \mathcal{A} \mapsto \mathbb{R}\}$ denote the set of all possible reward functions, and let $\mathcal{Q} = \{Q : \mathcal{S} \times \mathcal{A} \mapsto \mathbb{R}\}$ denote the set of all possible Q tables. Then, there exists a bijection mapping between $\mathcal{R}$ and $\mathcal{Q}$, induced by Bellman optimality equation.*

*Proof.* $\Rightarrow$ Given any reward function $R(s, a) \in \mathcal{R}$, define the Bellman operator as

$$H_R(Q)(s, a) = R(s, a) + \gamma \sum_{s'} P(s' \mid s, a) \max_{a'} Q(s', a'). \tag{39}$$

Since $\gamma < 1$, $H_R(Q)$ is a contraction mapping, i.e., $\|H_R(Q_1) - H_R(Q_2)\|_\infty \leq \gamma \|Q_1 - Q_2\|_\infty$, $\forall Q_1, Q_2 \in \mathcal{Q}$. Then by Banach Fixed Point Theorem, there is a unique $Q \in \mathcal{Q}$ that satisfies $Q = H_R(Q)$, which is the $Q$ that $R$ maps to.

$\Leftarrow$ Given any $Q \in \mathcal{Q}$, one can define the corresponding $R \in \mathcal{R}$ by

$$R(s, a) = Q(s, a) - \gamma \sum_{s'} P(s' \mid s, a) \max_{a'} Q(s', a'). \tag{40}$$

Thus the mapping is one-to-one. $\blacksquare$

**Proposition 1.** *The attack problem* (12)-(15) *is always feasible for any target policy $\pi^\dagger$.*

*Proof.* For any target policy $\pi^\dagger : \mathcal{S} \mapsto \mathcal{A}$, we construct the following $Q$:

$$Q(s, a) = \begin{cases} \varepsilon & \forall s \in \mathcal{S}, a = \pi^\dagger(s), \\ 0, & \text{otherwise.} \end{cases} \tag{41}$$

The $Q$ values in (41) satisfy the constraint (15). Note that we construct the $Q$ values so that for all $s \in \mathcal{S}$, $\max_a Q(s, a) = \varepsilon$. By proposition 4, the corresponding reward function induced by Bellman optimality equation is

$$\hat{R}(s, a) = \begin{cases} (1 - \gamma)\varepsilon & \forall s \in \mathcal{S}, a = \pi^\dagger(s), \\ -\gamma\varepsilon, & \text{otherwise.} \end{cases} \tag{42}$$

Then one can let $r_t = \hat{R}(s_t, a_t)$ so that $\mathbf{r} = (r_0, ..., r_{T-1})$, $\hat{R}$ in (42), together with $Q$ in (41) is a feasible solution to (12)-(15). $\blacksquare$

## B  Proof of Theorem 2

The proof of Theorem 2 relies on a few lemmas. We first prove the following result, which shows that given two vectors that have equal element summation, the vector whose elements are smoother will have smaller $\ell_\alpha$ norm for any $\alpha \geq 1$. This result is used later to prove Lemma 6.

**Lemma 5.** *Let $x, y \in \mathbb{R}^T$ be two vectors. Let $\mathcal{I} \subset \{0, 1, ..., T - 1\}$ be a subset of indexes such that*

$$i). \quad x_i = \frac{1}{|\mathcal{I}|} \sum_{j \in \mathcal{I}} y_j, \forall i \in \mathcal{I}, \qquad ii). \quad x_i = y_i, \forall i \neq \mathcal{I}. \tag{43}$$

*Then for any $\alpha \geq 1$, we have $\|x\|_\alpha \leq \|y\|_\alpha$.*

*Proof.* Note that the conditions $i)$ and $ii)$ suggest the summation of elements in $x$ and $y$ are equal, and only elements in $\mathcal{I}$ differ for the two vectors. However, the elements in $\mathcal{I}$ of $x$ are smoother than that of $y$, thus $x$ has smaller norm. To prove the result, we consider three cases separately.

Case 1: $\alpha = 1$. Then we have

$$\|x\|_\alpha - \|y\|_\alpha = \sum_i |x_i| - \sum_j |y_j| = \sum_{i \in \mathcal{I}} |x_i| - \sum_{j \in \mathcal{I}} |y_j| = |\sum_{j \in \mathcal{I}} y_j| - \sum_{j \in \mathcal{I}} |y_j| \le 0. \qquad (44)$$

Case 2: $1 < \alpha < \infty$. We show $\|x\|_\alpha^\alpha \le \|y\|_\alpha^\alpha$. Note that

$$\|x\|_\alpha^\alpha - \|y\|_\alpha^\alpha = \sum_i |x_i|^\alpha - \sum_j |y_j|^\alpha = \sum_{i \in \mathcal{I}} |x_i|^\alpha - \sum_{j \in \mathcal{I}} |y_j|^\alpha$$

$$= \frac{1}{|\mathcal{I}|^{\alpha-1}} |\sum_{j \in \mathcal{I}} y_j|^\alpha - \sum_{j \in \mathcal{I}} |y_j|^\alpha \le \frac{1}{|\mathcal{I}|^{\alpha-1}} (\sum_{j \in \mathcal{I}} |y_j|)^\alpha - \sum_{j \in \mathcal{I}} |y_j|^\alpha. \qquad (45)$$

Let $\beta = \frac{\alpha}{\alpha-1}$. By Holder's inequality, we have

$$\sum_{j \in \mathcal{I}} |y_j| \le (\sum_{j \in \mathcal{I}} |y_j|^\alpha)^{\frac{1}{\alpha}} (\sum_{j \in \mathcal{I}} 1^\beta)^{\frac{1}{\beta}} = (\sum_{j \in \mathcal{I}} |y_j|^\alpha)^{\frac{1}{\alpha}} |\mathcal{I}|^{1-\frac{1}{\alpha}}. \qquad (46)$$

Plugging (46) into (45), we have

$$\|x\|_\alpha^\alpha - \|y\|_\alpha^\alpha \le \frac{1}{|\mathcal{I}|^{\alpha-1}} (\sum_{j \in \mathcal{I}} |y_j|^\alpha) |\mathcal{I}|^{\alpha-1} - \sum_{j \in \mathcal{I}} |y_j|^\alpha = 0. \qquad (47)$$

Case 3: $\alpha = \infty$. We have

$$\|x\|_\alpha = \max_i |x_i| = \max\{\frac{1}{|\mathcal{I}|} |\sum_{j \in \mathcal{I}} y_j|, \max_{i \notin \mathcal{I}} |x_i|\} \le \max\{\frac{1}{|\mathcal{I}|} \sum_{j \in \mathcal{I}} |y_j|, \max_{i \notin \mathcal{I}} |x_i|\}$$

$$\le \max\{\max_{j \in \mathcal{I}} |y_j|, \max_{i \notin \mathcal{I}} |x_i|\} = \max\{\max_{j \in \mathcal{I}} |y_j|, \max_{j \notin \mathcal{I}} |y_j|\} = \max_j |y_j| = \|y\|_\alpha. \qquad (48)$$

Therefore $\forall \alpha \ge 1$, we have $\|x\|_\alpha \le \|y\|_\alpha$. ∎

Next we prove Lemma 6, which shows that one possible optimal attack solution to (12)-(15) takes the following form: shift all the clean rewards in $T_{s,a}$ by the same amount $\psi(s, a)$. Here $\psi(s, a)$ is a function of state $s$ and action $a$. That means, rewards belonging to different $T_{s,a}$ might be shifted a different amount, but those corresponding to the same $(s, a)$ pair will be identically shifted.

**Lemma 6.** *There exists a function $\psi(s, a)$ such that $r_t = r_t^0 + \psi(s_t, a_t)$, together with some $\hat{R}$ and $Q$, is an optimal solution to our attack problem (12)-(15).*

We point out that although there exists an optimal attack taking the above form, it is not necessarily the only optimal solution. However, all those optimal solutions must have exactly the same objective value (attack cost), thus it suffices to consider the solution in Lemma 6.

*Proof.* Let $\mathbf{r}^* = (r_0^*, ..., r_{T-1}^*)$, $\hat{R}^*$ and $Q^*$ be any optimal solution to (12)-(15). Fix a particular state-action pair $(s, a)$, we have

$$\hat{R}^*(s, a) = \frac{1}{|T_{s,a}|} \sum_{t \in T_{s,a}} r_t^*. \qquad (49)$$

Let $\hat{R}^0(s, a) = \frac{1}{|T_{s,a}|} \sum_{t \in T_{s,a}} r_t^0$ be the reward function for the $(s, a)$ pair estimated from clean data $\mathbf{r}^0$. We then define a different poisoned reward vector $\mathbf{r}' = (r_0', ..., r_{T-1}')$, where

$$r_t' = \begin{cases} r_t^0 + \hat{R}^*(s, a) - \hat{R}^0(s, a), & t \in T_{s,a}, \\ r_t^*, & t \notin T_{s,a}. \end{cases} \qquad (50)$$

Now we show $\mathbf{r}'$, $\hat{R}^*$ and $Q^*$ is another optimal solution to (12)-(15). We first verify that $\mathbf{r}'$, $\hat{R}^*$, and $Q^*$ satisfy constraints (13)-(15). To verify (13), we only need to check $\hat{R}^*(s, a) = \frac{1}{|T_{s,a}|} \sum_{t \in T_{s,a}} r_t'$, since $\mathbf{r}'$ and $\mathbf{r}^*$ only differ on those rewards in $T_{s,a}$. We have

$$\frac{1}{|T_{s,a}|} \sum_{t \in T_{s,a}} r_t' = \frac{1}{|T_{s,a}|} \sum_{t \in T_{s,a}} \left( r_t^0 + \hat{R}^*(s, a) - \hat{R}^0(s, a) \right)$$

$$= \hat{R}^0(s, a) + \hat{R}^*(s, a) - \hat{R}^0(s, a) = \hat{R}^*(s, a), \qquad (51)$$

Thus $\mathbf{r}'$ and $\hat{R}^*$ satisfy constraint (13). $\hat{R}^*$ and $Q^*$ obviously satisfy constraints (14) and (15) because $\mathbf{r}^*$, $\hat{R}^*$ and $Q^*$ is an optimal solution.

Let $\delta' = \mathbf{r}' - \mathbf{r}^0$ and $\delta^* = \mathbf{r}^* - \mathbf{r}^0$, then one can easily show that $\delta'$ and $\delta^*$ satisfy the conditions in Lemma 5 with $\mathcal{I} = T_{s,a}$. Therefore by Lemma 5, we have

$$\|\mathbf{r}' - \mathbf{r}^0\|_\alpha = \|\delta'\|_\alpha \leq \|\delta^*\|_\alpha = \|\mathbf{r}^* - \mathbf{r}^0\|_\alpha. \tag{52}$$

But note that by our assumption, $\mathbf{r}^*$ is an optimal solution, thus $\|\mathbf{r}^* - \mathbf{r}^0\|_\alpha \leq \|\mathbf{r}' - \mathbf{r}^0\|_\alpha$, which gives $\|\mathbf{r}' - \mathbf{r}^0\|_\alpha = \|\mathbf{r}^* - \mathbf{r}^0\|_\alpha$. This suggests $\mathbf{r}'$, $\hat{R}^*$, and $Q^*$ is another optimal solution. Compared to $\mathbf{r}^*$, $\mathbf{r}'$ differs in that $r_t' - r_t^0$ now becomes identical for all $t \in T_{s,a}$ for a particular $(s, a)$ pair. Reusing the above argument iteratively, one can make $r_t' - r_t^0$ identical for all $t \in T_{s,a}$ for all $(s, a)$ pairs, while guaranteeing the solution is still optimal. Therefore, we have

$$r_t' = r_t^0 + \hat{R}^*(s, a) - \hat{R}^0(s, a), \forall t \in T_{s,a}, \forall s, a, \tag{53}$$

together with $\hat{R}^*$ and $Q^*$ is an optimal solution to (12)-(15). Let $\psi(s, a) = \hat{R}^*(s, a) - \hat{R}^0(s, a)$ conclude the proof. ∎

Finally, Lemma 7 provides a sensitive analysis on the value function $Q$ as the reward function changes.

**Lemma 7.** *Let $\hat{M} = (\mathcal{S}, \mathcal{A}, \hat{P}, \hat{R}', \gamma)$ and $\hat{M}^0 = (\mathcal{S}, \mathcal{A}, \hat{P}, \hat{R}^0, \gamma)$ be two MDPs, where only the reward function differs. Let $Q'$ and $Q^0$ be action values satisfying the Bellman optimality equation on $\hat{M}$ and $\hat{M}^0$ respectively, then*

$$(1 - \gamma)\|Q' - Q^0\|_\infty \leq \|\hat{R} - \hat{R}^0\|_\infty \leq (1 + \gamma)\|Q' - Q^0\|_\infty. \tag{54}$$

*Proof.* Define the Bellman operator as

$$H_{\hat{R}}(Q)(s, a) = \hat{R}(s, a) + \gamma \sum_{s'} \hat{P}(s' \mid s, a) \max_{a'} Q(s', a'). \tag{55}$$

From now on we suppress variables $s$ and $a$ for convenience. Note that due to the Bellman optimality, we have $H_{\hat{R}^0}(Q^0) = Q^0$ and $H_{\hat{R}'}(Q') = Q'$, thus

$$\begin{aligned}
\|Q' - Q^0\|_\infty &= \|H_{\hat{R}'}(Q') - H_{\hat{R}^0}(Q^0)\|_\infty \\
&= \|H_{\hat{R}'}(Q') - H_{\hat{R}'}(Q^0) + H_{\hat{R}'}(Q^0) - H_{\hat{R}^0}(Q^0)\|_\infty \\
&\leq \|H_{\hat{R}'}(Q') - H_{\hat{R}'}(Q^0)\|_\infty + \|H_{\hat{R}'}(Q^0) - H_{\hat{R}^0}(Q^0)\|_\infty \\
&\leq \gamma\|Q' - Q^0\|_\infty + \|H_{\hat{R}'}(Q^0) - H_{\hat{R}^0}(Q^0)\|_\infty \text{ (by contraction of } H_{\hat{R}'}(\cdot)) \\
&= \gamma\|Q' - Q^0\|_\infty + \|\hat{R}' - \hat{R}^0\|_\infty \text{ (by } H_{\hat{R}'}(Q^0) - H_{\hat{R}^0}(Q^0) = \hat{R}' - \hat{R}^0)
\end{aligned} \tag{56}$$

Rearranging we have $(1 - \gamma)\|Q' - Q^0\|_\infty \leq \|\hat{R}' - \hat{R}^0\|_\infty$. Similarly we have

$$\begin{aligned}
\|Q' - Q^0\|_\infty &= \|H_{\hat{R}'}(Q') - H_{\hat{R}^0}(Q^0)\|_\infty \\
&= \|H_{\hat{R}'}(Q^0) - H_{\hat{R}^0}(Q^0) + H_{\hat{R}'}(Q') - H_{\hat{R}'}(Q^0)\|_\infty \\
&\geq \|H_{\hat{R}'}(Q^0) - H_{\hat{R}^0}(Q^0)\|_\infty - \|H_{\hat{R}'}(Q') - H_{\hat{R}'}(Q^0)\|_\infty \\
&\geq \|H_{\hat{R}'}(Q^0) - H_{\hat{R}^0}(Q^0)\|_\infty - \gamma\|Q' - Q^0\|_\infty \\
&= \|\hat{R}' - \hat{R}^0\|_\infty - \gamma\|Q' - Q^0\|_\infty
\end{aligned} \tag{57}$$

Rearranging we have $\|\hat{R}' - \hat{R}^0\|_\infty \leq (1 + \gamma)\|Q' - Q^0\|_\infty$, concluding the proof. ∎

Now we are ready to prove our main result.

**Theorem 2.** *Assume $\alpha \geq 1$ in (12). Let $\mathbf{r}^*$, $\hat{R}^*$ and $Q^*$ be an optimal solution to (12)-(15), then*

$$\frac{1}{2}(1 - \gamma)\Delta(\varepsilon) \left( \min_{s,a} |T_{s,a}| \right)^{\frac{1}{\alpha}} \leq \|\mathbf{r}^* - \mathbf{r}^0\|_\alpha \leq \frac{1}{2}(1 + \gamma)\Delta(\varepsilon)T^{\frac{1}{\alpha}}. \tag{16}$$

*Proof.* We construct the following value function $Q'$.

$$Q'(s, a) = \begin{cases} Q^0(s, a) + \dfrac{\Delta(\varepsilon)}{2}, & \forall s \in \mathcal{S}, a = \pi^\dagger(s), \\ Q^0(s, a) - \dfrac{\Delta(\varepsilon)}{2}, & \forall s \in \mathcal{S}, \forall a \neq \pi^\dagger(s). \end{cases} \tag{58}$$

Note that $\forall s \in \mathcal{S}$ and $\forall a \neq \pi^\dagger(s)$, we have

$$\begin{aligned} \Delta(\varepsilon) &= \max_{s' \in \mathcal{S}} [\max_{a' \neq \pi^\dagger(s')} Q^0(s', a') - Q^0(s', \pi^\dagger(s')) + \varepsilon]_+ \\ &\geq \max_{a' \neq \pi^\dagger(s)} Q^0(s, a') - Q^0(s, \pi^\dagger(s)) + \varepsilon \geq Q^0(s, a) - Q^0(s, \pi^\dagger(s)) + \varepsilon, \end{aligned} \tag{59}$$

which leads to

$$Q^0(s, a) - Q^0(s, \pi^\dagger(s)) - \Delta(\varepsilon) \leq -\varepsilon, \tag{60}$$

thus we have $\forall s \in \mathcal{S}$ and $\forall a \neq \pi^\dagger(s)$,

$$\begin{aligned} Q'(s, \pi^\dagger(s)) &= Q^0(s, \pi^\dagger(s)) + \frac{\Delta(\varepsilon)}{2} \\ &= Q^0(s, a) - [Q^0(s, a) - Q^0(s, \pi^\dagger(s)) - \Delta(\varepsilon)] - \frac{\Delta(\varepsilon)}{2} \\ &\geq Q^0(s, a) + \varepsilon - \frac{\Delta(\varepsilon)}{2} = Q'(s, a) + \varepsilon. \end{aligned} \tag{61}$$

Therefore $Q'$ satisfies the constraint (15). By proposition 4, there exists a unique function $R'$ such that $Q'$ satisfies the Bellman optimality equation of MDP $\hat{M}' = (\mathcal{S}, \mathcal{A}, \hat{P}, R', \gamma)$. We then construct the following reward vector $\mathbf{r}' = (r'_0, ..., r'_{T-1})$ such that $\forall (s, a)$ and $\forall t \in T_{s,a}$, $r'_t = r^0_t + R'(s, a) - \hat{R}^0(s, a)$, where $\hat{R}^0(s, a)$ is the reward function estimated from $\mathbf{r}^0$. The reward function estimated on $\mathbf{r}'$ is then

$$\begin{aligned} \hat{R}'(s, a) &= \frac{1}{|T_{s,a}|} \sum_{t \in T_{s,a}} r'_t = \frac{1}{|T_{s,a}|} \sum_{t \in T_{s,a}} \left( r^0_t + R'(s, a) - \hat{R}^0(s, a) \right) \\ &= \hat{R}^0(s, a) + R'(s, a) - \hat{R}^0(s, a) = R'(s, a). \end{aligned} \tag{62}$$

Thus $\mathbf{r}'$, $\hat{R}'$ and $Q'$ is a feasible solution to (12)-(15). Now we analyze the attack cost for $\mathbf{r}'$, which gives us a natural upper bound on the attack cost of the optimal solution $\mathbf{r}^*$. Note that $Q'$ and $Q^0$ satisfy the Bellman optimality equation for reward function $\hat{R}'$ and $\hat{R}^0$ respectively, and

$$\|Q' - Q^0\|_\infty = \frac{\Delta(\varepsilon)}{2}, \tag{63}$$

thus by Lemma 7, we have $\forall t$,

$$\begin{aligned} |r'_t - r^0_t| = |\hat{R}'(s_t, a_t) - \hat{R}^0(s_t, a_t)| &\leq \max_{s,a} |\hat{R}'(s, a) - \hat{R}^0(s, a)| = \|\hat{R}' - \hat{R}^0\|_\infty \\ &\leq (1 + \gamma)\|Q' - Q^0\|_\infty = \frac{1}{2}(1 + \gamma)\Delta(\varepsilon). \end{aligned} \tag{64}$$

Therefore, we have

$$\|\mathbf{r}^* - \mathbf{r}^0\|_\alpha \leq \|\mathbf{r}' - \mathbf{r}^0\|_\alpha = (\sum_{t=0}^{T-1} |r'_t - r^0_t|^\alpha)^{\frac{1}{\alpha}} \leq \frac{1}{2}(1 + \gamma)\Delta(\varepsilon)T^{\frac{1}{\alpha}}. \tag{65}$$

Now we prove the lower bound. We consider two cases separately.

Case 1: $\Delta(\varepsilon) = 0$. We must have $Q^0(s, \pi^\dagger(s)) \geq Q^0(s, a) + \varepsilon$, $\forall s \in \mathcal{S}, \forall a \neq \pi^\dagger(s)$. In this case no attack is needed and therefore the optimal solution is $\mathbf{r}^* = \mathbf{r}^0$. The lower bound holds trivially.

Case 2: $\Delta(\varepsilon) > 0$. Let $s'$ and $a'$ ($a' \neq \pi^\dagger(s')$) be a state-action pair such that

$$\Delta(\varepsilon) = Q^0(s', a') - Q^0(s', \pi^\dagger(s')) + \varepsilon. \tag{66}$$

Let $\mathbf{r}^*$, $\hat{R}^*$ and $Q^*$ be an optimal solution to (12)-(15) that takes the form in Lemma 6, i.e.,

$$r_t^* = r_t^0 + \hat{R}^*(s,a) - \hat{R}^0(s,a), \forall t \in T_{s,a}, \forall s, a. \tag{67}$$

Constraint (15) ensures that $Q^*(s', \pi^\dagger(s')) \geq Q^*(s', a') + \varepsilon$, in which case either one of the following two conditions must hold:

$$i). \quad Q^*(s', \pi^\dagger(s')) - Q^0(s', \pi^\dagger(s')) \geq \frac{\Delta(\varepsilon)}{2}, \qquad ii). \quad Q^0(s', a') - Q^*(s', a') \geq \frac{\Delta(\varepsilon)}{2}, \tag{68}$$

since otherwise we have

$$Q^*(s', \pi^\dagger(s')) < Q^0(s', \pi^\dagger(s')) + \frac{\Delta(\varepsilon)}{2} = Q^0(s', \pi^\dagger(s')) + \frac{1}{2}[Q^0(s', a') - Q^0(s', \pi^\dagger(s')) + \varepsilon]$$

$$= \frac{1}{2}Q^0(s', a') + \frac{1}{2}Q^0(s', \pi^\dagger(s')) + \frac{\varepsilon}{2} = Q^0(s', a') - \frac{1}{2}[Q^0(s', a') - Q^0(s', \pi^\dagger(s')) + \varepsilon] + \varepsilon$$

$$= Q^0(s', a') - \frac{\Delta(\varepsilon)}{2} + \varepsilon < Q^*(s', a') + \varepsilon. \tag{69}$$

Next note that if either $i$) or $ii$) holds, we have $\|Q^* - Q^0\|_\infty \geq \frac{\Delta(\varepsilon)}{2}$. By Lemma 7, we have

$$\max_{s,a} |\hat{R}^*(s,a) - \hat{R}^0(s,a)| = \|\hat{R}^* - \hat{R}^0\|_\infty \geq (1-\gamma)\|Q^* - Q^0\|_\infty \geq \frac{1}{2}(1-\gamma)\Delta(\varepsilon). \tag{70}$$

Let $s^*, a^* \in \arg\max_{s,a} |\hat{R}^*(s,a) - \hat{R}^0(s,a)|$, then we have

$$|\hat{R}^*(s^*, a^*) - \hat{R}^0(s^*, a^*)| \geq \frac{1}{2}(1-\gamma)\Delta(\varepsilon). \tag{71}$$

Therefore, we have

$$\|\mathbf{r}^* - \mathbf{r}^0\|_\alpha^\alpha = \sum_{t=0}^{T-1} |r_t^* - r_t^0|^\alpha = \sum_{s,a} \sum_{t \in T_{s,a}} |r_t^* - r_t^0|^\alpha \geq \sum_{t \in T_{s^*,a^*}} |r_t^* - r_t^0|^\alpha$$

$$= \sum_{t \in T_{s^*,a^*}} |\hat{R}^*(s^*, a^*) - \hat{R}^0(s^*, a^*)|^\alpha \geq \left(\frac{1}{2}(1-\gamma)\Delta(\varepsilon)\right)^\alpha |T_{s^*,a^*}| \tag{72}$$

$$\geq \left(\frac{1}{2}(1-\gamma)\Delta(\varepsilon)\right)^\alpha \min_{s,a} |T_{s,a}|.$$

Therefore $\|\mathbf{r}^* - \mathbf{r}^0\|_\alpha \geq \frac{1}{2}(1-\gamma)\Delta(\varepsilon) (\min_{s,a} |T_{s,a}|)^{\frac{1}{\alpha}}$.

We finally point out that while an optimal solution $\mathbf{r}^*$ may not necessarily take the form in Lemma 6, it suffices to bound the cost of an optimal attack which indeed takes this form (as we did in the proof) since all optimal attacks have exactly the same objective value. ∎

## C   Convex Surrogate for LQR Attack Optimization

By pulling the positive semi-definite constraints on $Q$ and $R$ out of the lower level optimization (32), one can turn the original attack optimization (27)-(33) into the following surrogate optimization:

$$\min_{\mathbf{r},\hat{Q},\hat{R},\hat{q},\hat{c},X,x} \quad \|\mathbf{r} - \mathbf{r}_0\|_\alpha \tag{73}$$

$$\text{s.t.} \quad -\gamma\left(\hat{R} + \gamma\hat{B}^\top X\hat{B}\right)^{-1}\hat{B}^\top X\hat{A} = K^\dagger, \tag{74}$$

$$-\gamma\left(\hat{R} + \gamma\hat{B}^\top X\hat{B}\right)^{-1}\hat{B}^\top x = k^\dagger, \tag{75}$$

$$X = \gamma\hat{A}^\top X\hat{A} - \gamma^2\hat{A}^\top X\hat{B}\left(\hat{R} + \gamma\hat{B}^\top X\hat{B}\right)^{-1}\hat{B}^\top X\hat{A} + \hat{Q} \tag{76}$$

$$x = \hat{q} + \gamma(\hat{A} + \hat{B}K^\dagger)^\top x \tag{77}$$

$$(\hat{Q}, \hat{R}, \hat{q}, \hat{c}) = \arg\min \sum_{t=0}^{T-1} \left\|\frac{1}{2}s_t^\top Q s_t + q^\top s_t + a_t^\top R a_t + c + r_t\right\|_2^2 \tag{78}$$

$$\hat{Q} \succeq 0, \hat{R} \succeq \varepsilon I, X \succeq 0. \tag{79}$$

The feasible set of (73)-(79) is a subset of the original problem, thus the surrogate attack optimization is a more stringent formulation than the original attack optimization, that is, successfully solving the surrogate optimization gives us a (potentially) sub-optimal solution to the original problem. To see why the surrogate optimization is more stringent, we illustrate with a much simpler example as below. A formal proof is straight forward, thus we omit it here. The original problem is (80)-(81). The feasible set for $\hat{a}$ is a singleton set $\{0\}$, and the optimal objective value is 0.

$$\min_{\hat{a}} \quad 0 \tag{80}$$

$$\text{s.t.} \quad \hat{a} = \arg\min_{a \geq 0}(a+3)^2, \tag{81}$$

Once we pull the constraint out of the lower-level optimization (81), we end up with a surrogate optimization (82)-(84). Note that (83) requires $\hat{a} = -3$, which does not satisfy (84). Therefore the feasible set of the surrogate optimization is $\emptyset$, meaning it is more stringent than (80)-(81).

$$\min_{\hat{a}} \quad 0 \tag{82}$$

$$\text{s.t.} \quad \hat{a} = \arg\min(a+3)^2, \tag{83}$$

$$\hat{a} \geq 0 \tag{84}$$

Back to our attack optimization (73)-(79), this surrogate attack optimization comes with the advantage of being convex, thus can be solved to global optimality.

**Proposition 8.** *The surrogate attack optimization* (73)-(79) *is convex.*

*Proof.* First note that the sub-level optimization (78) is itself a convex problem, thus is equivalent to the corresponding KKT condition. We write out the KKT condition of (78) to derive an explicit form of our attack formulation as below:

$$\min_{\mathbf{r},\hat{Q},\hat{R},\hat{q},\hat{c},X,x} \quad \|\mathbf{r} - \mathbf{r}_0\|_\alpha \tag{85}$$

$$\text{s.t.} \quad -\gamma \left(\hat{R} + \gamma \hat{B}^\top X \hat{B}\right)^{-1} \hat{B}^\top X \hat{A} = K^\dagger, \tag{86}$$

$$-\gamma \left(\hat{R} + \gamma \hat{B}^\top X \hat{B}\right)^{-1} \hat{B}^\top x = k^\dagger, \tag{87}$$

$$X = \gamma \hat{A}^\top X \hat{A} - \gamma^2 \hat{A}^\top X \hat{B} \left(\hat{R} + \gamma \hat{B}^\top X \hat{B}\right)^{-1} \hat{B}^\top X \hat{A} + \hat{Q} \tag{88}$$

$$x = \hat{q} + \gamma(\hat{A} + \hat{B}K^\dagger)^\top x \tag{89}$$

$$\sum_{t=0}^{T-1}(\frac{1}{2}s_t^\top \hat{Q}s_t + \hat{q}^\top s_t + a_t^\top \hat{R}a_t + \hat{c} + r_t)s_t s_t^\top = 0, \tag{90}$$

$$\sum_{t=0}^{T-1}(\frac{1}{2}s_t^\top \hat{Q}s_t + \hat{q}^\top s_t + a_t^\top \hat{R}a_t + \hat{c} + r_t)a_t a_t^\top = 0, \tag{91}$$

$$\sum_{t=0}^{T-1}(\frac{1}{2}s_t^\top \hat{Q}s_t + \hat{q}^\top s_t + a_t^\top \hat{R}a_t + \hat{c} + r_t)s_t = 0, \tag{92}$$

$$\sum_{t=0}^{T-1}(\frac{1}{2}s_t^\top \hat{Q}s_t + \hat{q}^\top s_t + a_t^\top \hat{R}a_t + \hat{c} + r_t) = 0, \tag{93}$$

$$\hat{Q} \succeq 0, \hat{R} \succeq \varepsilon I, X \succeq 0. \tag{94}$$

The objective is obviously convex. (86)-(88) are equivalent to

$$-\gamma \hat{B}^\top X \hat{A} = \left(\hat{R} + \gamma \hat{B}^\top X \hat{B}\right) K^\dagger. \tag{95}$$

$$-\gamma \hat{B}^\top x = \left(\hat{R} + \gamma \hat{B}^\top X \hat{B}\right) k^\dagger. \tag{96}$$

$$X = \gamma \hat{A}^\top X(\hat{A} + \hat{B}K^\dagger) + \hat{Q}, \tag{97}$$

Note that these three equality constraints are all linear in $X$, $\hat{R}$, $x$, and $\hat{Q}$. (89) is linear in $x$ and $\hat{q}$. (90)-(93) are also linear in $\hat{Q}$, $\hat{R}$, $\hat{q}$, $\hat{c}$ and $\mathbf{r}$. Finally, (94) contains convex constraints on $\hat{Q}$, $\hat{R}$, and $X$. Given all above, the attack problem is convex. ∎

Next we analyze the feasibility of the surrogate attack optimization.

**Proposition 9.** *Let $\hat{A}$, $\hat{B}$ be the learner's estimated transition kernel. Let*

$$L^{\dagger}(s,a) = \frac{1}{2}s^{\top}Q^{\dagger}s + (q^{\dagger})^{\top}s + a^{\top}R^{\dagger}a + c^{\dagger} \tag{98}$$

*be the attacker-defined loss function. Assume $R^{\dagger} \succeq \varepsilon I$. If the target policy $K^{\dagger}$, $k^{\dagger}$ is the optimal control policy induced by the LQR with transition kernel $\hat{A}$, $\hat{B}$, and loss function $L^{\dagger}(s,a)$, then the surrogate attack optimization (73)-(79) is feasible. Furthermore, the optimal solution can be achieved.*

*Proof.* To prove feasibility, it suffices to construct a feasible solution to optimization (73)-(79). Let

$$r_t = \frac{1}{2}s_t^{\top}Q^{\dagger}s_t + {q^{\dagger}}^{\top}s_t + a_t^{\top}R^{\dagger}a_t + c^{\dagger} \tag{99}$$

and $\mathbf{r}$ be the vector whose $t$-th element is $r_t$. We next show that $\mathbf{r}$, $Q^{\dagger}$, $R^{\dagger}$, $q^{\dagger}$, $c^{\dagger}$, together with some $X$ and $x$ is a feasible solution. Note that since $K^{\dagger}$, $k^{\dagger}$ is induced by the LQR with transition kernel $\hat{A}$, $\hat{B}$ and cost function $L^{\dagger}(s,a)$, constraints (74)-(77) must be satisfied with some $X$ and $x$. The poisoned reward vector $\mathbf{r}$ obviously satisfies (78) since it is constructed exactly as the minimizer. By our assumption, $R^{\dagger} \succeq \varepsilon I$, thus (79) is satisfied. Therefore, $\mathbf{r}$, $Q^{\dagger}$, $R^{\dagger}$, $q^{\dagger}$, $c^{\dagger}$, together with the corresponding $X$, $x$ is a feasible solution, and the optimization (73)-(79) is feasible. Furthermore, since the feasible set is closed, the optimal solution can be achieved. ∎

# D    Conditions for The LQR Learner to Have Unique Estimate

The LQR learner estimates the cost function by

$$(\hat{Q}, \hat{R}, \hat{q}, \hat{c}) = \underset{(Q \succeq 0, R \succeq \varepsilon I, q, c)}{\arg\min} \frac{1}{2}\sum_{t=0}^{T-1}\left\|\frac{1}{2}s_t^{\top}Qs_t + q^{\top}s_t + a_t^{\top}Ra_t + c + r_t\right\|_2^2. \tag{100}$$

We want to find a condition that guarantees the uniqueness of the solution.

Let $\psi \in \mathbb{R}^T$ be a vector, whose $t$-th element is

$$\psi_t = \frac{1}{2}s_t^{\top}Qs_t + q^{\top}s_t + a_t^{\top}Ra_t + c, 0 \le t \le T-1. \tag{101}$$

Note that we can view $\psi$ as a function of $D$, $Q$, $R$, $q$, and $c$, thus we can also denote $\psi(D, Q, R, q, c)$. Define $\Psi(D) = \{\psi(D, Q, R, q, c) \mid Q \succeq 0, R \succeq \varepsilon I, q, c\}$, i.e., all possible vectors that are achievable with form (101) if we vary $Q$, $R$, $q$ and $c$ subject to positive semi-definite constraints on $Q$ and $R$. We can prove that $\Psi$ is a closed convex set.

**Proposition 10.** $\forall D$, $\Psi(D) = \{\psi(D, Q, R, q, c) \mid Q \succeq 0, R \succeq \varepsilon I, q, c\}$ *is a closed convex set.*

*Proof.* Let $\psi_1, \psi_2 \in \Psi(D)$. We use $\psi_{i,t}$ to denote the $t$-th element of vector $\psi_i$. Then we have

$$\psi_{1,t} = \frac{1}{2}s_t^{\top}Q_1s_t + q_1^{\top}s_t + a_t^{\top}R_1a_t + c_1 \tag{102}$$

for some $Q_1 \succeq 0$, $R_1 \succeq \varepsilon I$, $q_1$ and $c_1$, and

$$\psi_{2,t} = \frac{1}{2}s_t^{\top}Q_2s_t + q_2^{\top}s_t + a_t^{\top}R_2a_t + c_2 \tag{103}$$

for some $Q_2 \succeq 0$, $R_2 \succeq \varepsilon I$, $q_2$ and $c_2$. $\forall k \in [0,1]$, let $\psi_3 = k\psi_1 + (1-k)\psi_2$. Then the $t$-th element of $\psi_3$ is

$$\begin{aligned}\psi_{3,t} =& \frac{1}{2}s_t^{\top}[kQ_1 + (1-k)Q_2]s_t + [kq_1 + (1-k)q_2]^{\top}s_t \\ & + a_t^{\top}[kR_1 + (1-k)R_2]a_t + kc_1 + (1-k)c_2\end{aligned} \tag{104}$$

Since $kQ_1 + (1-k)Q_2 \succeq 0$ and $kR_1 + (1-k)R_2 \succeq \varepsilon I$, $\psi_3 \in \Psi(D)$, concluding the proof. ∎

The optimization (100) is intrinsically a least-squares problem with positive semi-definite constraints on $Q$ and $R$, and is equivalent to solving the following linear equation:

$$\frac{1}{2}s_t^\top \hat{Q} s_t + \hat{q}^\top s_t + a_t^\top \hat{R} a_t + \hat{c} = \psi_t^*, \forall t, \tag{105}$$

where $\psi^* = \arg\min_{\psi \in \Psi(D)} \|\psi + \mathbf{r}\|_2^2$ is the projection of the negative reward vector $-\mathbf{r}$ onto the set $\Psi(D)$. The solution to (105) is unique if and only if the following two conditions both hold

    $i)$. The projection $\psi^*$ is unique.

    $ii)$. (105) has a unique solution for $\psi^*$.

Condition $i)$ is satisfied because $\Psi(D)$ is convex, and any projection (in $\ell_2$ norm) onto a convex set exists and is always unique (see Hilbert Projection Theorem). We next analyze when condition $ii)$ holds. (105) is a linear function in $\hat{Q}$, $\hat{R}$, $\hat{q}$, and $\hat{c}$, thus one can vectorize $\hat{Q}$ and $\hat{R}$ to obtain a problem in the form of linear regression. Then the uniqueness is guaranteed if and only if the design matrix has full column rank. Specifically, let $\hat{Q} \in \mathbb{R}^{n \times n}$, $\hat{R} \in \mathbb{R}^{m \times m}$, and $\hat{q} \in \mathbb{R}^n$. Let $s_{t,i}$ and $a_{t,i}$ denote the $i$-th element of $s_t$ and $a_t$ respectively. Define

$$\mathbf{A} = \left[ \begin{array}{ccccc|ccccc|c|c}
\frac{s_{0,1}^2}{2} & \cdots & \frac{s_{0,i}s_{0,j}}{2} & \cdots & \frac{s_{0,n}^2}{2} & a_{0,1}^2 & \cdots & a_{0,i}a_{0,j} & \cdots & a_{0,m}^2 & s_0^\top & 1 \\
\frac{s_{1,1}^2}{2} & \cdots & \frac{s_{1,i}s_{1,j}}{2} & \cdots & \frac{s_{1,n}^2}{2} & a_{1,1}^2 & \cdots & a_{1,i}a_{2,j} & \cdots & a_{1,m}^2 & s_1^\top & 1 \\
\vdots & & \vdots & & \vdots & \vdots & & \vdots & & \vdots & \vdots & \vdots \\
\frac{s_{t,1}^2}{2} & \cdots & \frac{s_{t,i}s_{t,j}}{2} & \cdots & \frac{s_{t,n}^2}{2} & a_{t,1}^2 & \cdots & a_{t,i}a_{t,j} & \cdots & a_{t,m}^2 & s_t^\top & 1 \\
\vdots & & \vdots & & \vdots & \vdots & & \vdots & & \vdots & \vdots & \vdots \\
\frac{s_{T-1,1}^2}{2} & \cdots & \frac{s_{T-1,i}s_{T-1,j}}{2} & \cdots & \frac{s_{T-1,n}^2}{2} & a_{T-1,1}^2 & \cdots & a_{T-1,i}a_{T-1,j} & \cdots & a_{T-1,m}^2 & s_{T-1}^\top & 1
\end{array} \right],$$

$$\mathbf{x}^\top = \left[ \begin{array}{ccccc|ccccc|ccccc|c} \hat{Q}_{11} & \cdots & \hat{Q}_{ij} & \cdots & \hat{Q}_{nn} & \hat{R}_{11} & \cdots & \hat{R}_{ij} & \cdots & \hat{R}_{mm} & \hat{q}_1 & \cdots & \hat{q}_i & \cdots & \hat{q}_n & \hat{c} \end{array} \right],$$

then (105) is equivalent to $\mathbf{A}\mathbf{x} = \psi^*$, where $\mathbf{x}$ contains the vectorized variables $\hat{Q}$, $\hat{R}$, $\hat{q}$ and $\hat{c}$. $\mathbf{A}\mathbf{x} = \psi^*$ has a unique solution if and only if $\mathbf{A}$ has full column rank.

## E    Sparse Attacks on TCE and LQR

In this section, we present experimental details for both TCE and LQR victims when the attacker uses $\ell_1$ norm to measure the attack cost, i.e. $\alpha = 1$. The other experimental parameters are set exactly the same as in the main text.

We first show the result for MDP experiment 2 with $\alpha = 1$, see Figure 4. The attack cost is $\|\mathbf{r} - \mathbf{r}^0\|_1 = 3.27$, which is small compared to $\|\mathbf{r}^0\|_1 = 105$. We note that the reward poisoning is extremely sparse: only the reward corresponding to action "go up" at the terminal state $G$ is increased by 3.27, and all other rewards remain unchanged. To explain this attack, first note that we set the target action for the terminal state to "go up", thus the corresponding reward must be increased. Next note that after the attack, the terminal state becomes a sweet spot, where the agent can keep taking action "go up" to gain large amount of discounted future reward. However, such future reward is discounted more if the agent reaches the terminal state via a longer path. Therefore, the agent will choose to go along the red trajectory to get into the terminal state earlier, though at a price of two discounted $-10$ rewards.

The result is similar for MDP experiment 3. The attack cost is $\|\mathbf{r} - \mathbf{r}^0\|_1 = 1.05$, compared to $\|\mathbf{r}^0\|_1 = 121$. In Figure 5, we show the reward modification for each state action pair. Again, the attack is very sparse: only rewards of 12 state-action pairs are modified out of a total of 124.

Finally, we show the result on attacking LQR with $\alpha = 1$. The attack cost is $\|\mathbf{r} - \mathbf{r}^0\|_1 = 5.44$, compared to $\|\mathbf{r}^0\|_1 = 2088.57$. In Figure 6, we plot the clean and poisoned trajectory of the vehicle, together with the reward modification in each time step. The attack is as effective as with a dense 2-norm attack in Figure 3. However, the poisoning is highly sparse: only 10 out of 400 rewards are changed.

Figure 4: Sparse reward modification for MDP experiment 2.

(a) Clean and poisoned vehicle trajectory.

(b) Clean and poisoned rewards.

Figure 6: Sparse-poisoning a vehicle running LQR in 4D state space.

# F Derivation of Discounted Discrete-time Algebraic Riccati Equation

We provide a derivation for the discounted Discrete-time Algebraic Riccati Equation. For simplicity, we consider the noiseless case, but the derivation easily generalizes to noisy case. We consider the

Figure 5: Sparse reward modification for MDP experiment 3.

loss function is a general quadratic function w.r.t. $s$ as follows:

$$L(s,a) = \frac{1}{2}s^\top Q s + q^\top s + c + a^\top R a. \tag{106}$$

When $q = 0, c = 0$, we recover the classic LQR setting. Assume the general value function takes the form $V(s) = \frac{1}{2}s^\top X s + s^\top x + v$. Let $Q(s,a)$ (note that this is different notation from the $Q$ matrix in $L(s,a)$) be the corresponding action value function. We perform dynamics programming as follows:

$$
\begin{aligned}
Q(s,a) &= \frac{1}{2}s^\top Q s + q^\top s + c + a^\top R a + \gamma V(As + Ba) \\
&= \frac{1}{2}s^\top Q s + q^\top s + c + a^\top R a + \gamma \left( \frac{1}{2}(As + Ba)^\top X (As + Ba) + (As + Ba)^\top x + v \right) \\
&= \frac{1}{2}s^\top (Q + \gamma A^\top X A)s + \frac{1}{2}a^\top (R + \gamma B^\top X B)a + s^\top (\gamma A^\top X B)a \\
&\quad + s^\top (q + \gamma A^\top x) + a^\top (\gamma B^\top x) + (c + \gamma v).
\end{aligned}
\tag{107}
$$

We minimize $a$ above:

$$
\begin{aligned}
&(R + \gamma B^\top X B)a + \gamma B^\top X A s + \gamma B^\top x = 0 \\
&\Rightarrow a = -\gamma(R + \gamma B^\top X B)^{-1}B^\top X A s - \gamma(R + \gamma B^\top X B)^{-1}B^\top x \triangleq Ks + k.
\end{aligned}
\tag{108}
$$

Now we substitute it back to $Q(s, a)$ and regroup terms, we get:

$$V(s) = \frac{1}{2} s^\top (Q + \gamma A^\top X A + K^\top (R + \gamma B^\top X B) K + 2\gamma A^\top X B K) s$$
$$+ s^\top (K^\top (R + \gamma B^\top X B) k + \gamma A^\top X B k + q + \gamma A^\top x + \gamma K^\top B^\top x) + C \tag{109}$$

for some constant $C$, which gives us the following recursion:

$$X = \gamma A^\top X A - \gamma^2 A^\top X B (R + \gamma B^\top X B)^{-1} B^\top X A + Q,$$
$$x = q + \gamma (A + BK)^\top x. \tag{110}$$