[Reviews · NeurIPS 2019]

Reviewer 1



Originality: the work proposed in this paper looks rather original, since it seems to be the first time that a paper addresses the problem of data poisoning in batch reinforcement learning. Quality: the paper presents interesting ideas, and shows that the policy poisoning problem can be reformulated as a bi-level optimisation problem. Two derivations of this optimisation problem are proposed (TCE and LQR), for which the authors propose analyses regarding the feasibility and costs of potential attacks. The paper also proposes 4 sets of experiments demonstrating the fact that the attack can actually force the RL agent to learn a policy chosen by the attack. Clarity: the paper is quite well structured. Section 4.2 could be a bit more structured, for instance by proposing claims and proofs in order to better guide the reader in the argumentation. Significance: this is indeed a crucial topic. The paper seems to open interesting discussions regarding the problem of damaging data in order to influence the decision of a potential victim controller. *** Post feedback *** Thanks to the authors for the feedback and clarification. I have updated the overall score accordingly.

Reviewer 2



The paper is well written and organized. It discusses the possibilities of attacks on an RL system by just modifying the rewards, where the RL algorithms are table based Q function and linear quadratic regulator. I think, that it is pretty obvious, that it is possible to change the policy in any way, if one has full control over the rewards. The paper provides a couple of experiments in which the changes to the rewards look surprisingly small, like 12%. But this does not mean much. If I change the reward in only one state transition sufficiently much, all other rewards might not need changes, and so by dividing the one large change by the number of state transitions in the data gives a small relative change. In addition, I criticize the definition of the poisoning ratio ||r - r^0|| / ||r|| for two reasons, 1. if all rewards are increased by an offset of let's say 1e9, the policy will not change, but the poisoning ratio does, 2. it seems odd to divide by ||r||, r being the changed reward, more natural would be ||r^0||, the original reward. I think a definition for the poisoning ratio as ||r-r^0|| / ||r^0 - mean(r^0)||, would be a better measure. AFTER FEEDBACK I am pleased with the author's feedback. In addition, after reading the comments of reviewers 1 and 4, I think that I slightly under-estimated the importance of the topic. Therefore I increased the "overall score" to 6.

Reviewer 3



The paper studies the problem of policy poisoning in batch reinforcement learning and control where the learner estimates the model of the world from batch data set, and finds an optimal policy with respect to the learned model. The attacker modifies the data by the means of modifying the reward entries to make the agent learn a target policy. The paper presents a framework for solving batch policy poison attacks on two standard victims. The theoretical and experimental results show some evidence for the feasibility of policy poisoning attacks. Overall, I think this is an interesting paper that is motivated under a realistic adversarial setting where the attacker can alter the reward (instead of altering the dynamics of the world) to change the optimal policy to an adversarial target policy. The paper is easy to read due to its clear organization. Further I appreciated the source code made available with the submission, so thank you for doing this. However, I have two major issues with the paper which I hope that can be clarified by the authors. i) While I understand the overall idea of your bilevel programming model, it seems that you are not following a formal notation (i.e., the Bard et al. 2000 notation). Therefore it is not clear if you have two separate followers problems, what are the decision variables of (each?) follower etc. Further, it is confusing to use r, \mathbb{r}, R, \mathcal{R}, \tilde{R} etc. all in one model as the reader cannot tell what variables are controlled by the leader or the follower (and that is very important). Please clarify this in your rebuttal and the revised version of your paper. ii) The experimental results are underwhelming where the size of the problems are very small and the domain is limited to a grids world problem - which does not tie well with the realistic adversarial setting introduced earlier. Is this because solving the adversarial optimization problem is not scalable? If so, this is important to note in the paper. Further, have you experimented with other and/or larger domains? One piece of literature that can be of interest to the authors is “Data Poisoning Attacks against Autoregressive Models”. Minor correction: line 161 the word ‘itself’ is repeated twice **After Rebuttal** Thank you for your responses, they were helpful clarifying my questions. Therefore, I am increasing my final score. Please include your clarifications in the final version of the paper.

[Author Response · NeurIPS 2019]

**Response to all reviewers**:

We thank the reviewers for carefully reviewing our paper and providing constructive feedback. To the best of our knowledge, we are the first paper formulating policy poisoning in batch RL and control, and there are fruitful new directions to explore, such as extending our attack to more complex RL learners. We believe our paper will trigger a line of new research on data poisoning attack on RL/Control.

**Response to reviewer 1**: We thank reviewer 1 for positive comments.

We will move some of the analysis and proofs back to the main text for better readability.

**Response to reviewer 3**: We thank the reviewer for valuable comments.

*(1) Is the attack trivial?* Our study has an emphasis on finding the \*optimal\* way of changing the rewards to achieve a target policy. The optimality guarantee is not trivial - one needs to solve a (bi-level) optimization to achieve it. Furthermore, we provide theoretical analysis to bound the optimal change of rewards measured by $\ell_p$ norm for arbitrary $p$, which is important to understanding the minimal effort an attacker has to spend to achieve successful attack.

*(2) Definition of poisoning ratio.* We apologize for the confusing definition in the paper. In our experiment, we actually computed poisoning ratio as $\|\mathbf{r} - \mathbf{r}^0\|_2/\|\mathbf{r}_0\|_2$ instead of $\|\mathbf{r} - \mathbf{r}^0\|_2/\|\mathbf{r}\|_2$, so this is just a typo of writing. We agree with the reviewer that if all rewards are shifted by a constant, the policy does not change, but the poisoning ratio does. However, the poisoning ratio is a metric that tries to capture the notion of \*attack cost\*, which not only depends on the policy change, but also ties to the magnitude of the clean rewards. Conceptually, the same "absolute" change on larger clean rewards should mean smaller attack cost, thus should have smaller poisoning ratio. This is exactly the case if we divide by $\|r_0\|$. Besides, from the optimization perspective, our attack optimizes $\|\mathbf{r} - \mathbf{r}_0\|$, which is equivalent to optimizing $\|\mathbf{r} - \mathbf{r}_0\|/\|\mathbf{r}_0\|$ or $\|\mathbf{r} - \mathbf{r}_0\|/\|\mathbf{r}_0 - \text{mean}(\mathbf{r}_0)\|$. But we do think the reviewer's suggestion is also reasonable, so we computed the poisoning ratio using the suggested metric $\|\mathbf{r} - \mathbf{r}_0\|/\|\mathbf{r}_0 - \text{mean}(\mathbf{r}_0)\|$, and the results for experiment 2,3 and 4 are 14.66%, 8.64%, and 0.77% respectively, which is just slightly higher than those reported in the paper. We will include the metric suggested by the reviewer in the revised version.

Regarding large attack on a single transition, it is true that small poisoning ratio does not fully capture the magnitude of change on each individual reward, if the metric is $\ell_2$ norm (as we did in our experiments). However, as our formulation uses general $\ell_p$ norm, one can consider using $\ell_\infty$ norm instead, which will avoid the example the reviewer mentioned.

**Response to reviewer 4**: We thank the reviewer for constructive comments.

*(1) Bi-level optimization formulation.* We follow the notation in state-of-the-art literatures on data poisoning attacks (e.g, [1]). However, we can map the notation to Bard et al. 2000 paper as follows.

The leader optimization is (25), where the decision variables is the poisoned rewards $\mathbf{r}$. We have only one follower problem (30), together with LQR solution constraint (26) to (29). The decision variables for the follower problem is on the LHS of (30). The (19) is not a follower problem because the estimates $\hat{A}$ and $\hat{B}$ are attack-independent, thus can be computed beforehand based on the clean data. The estimates $\hat{A}$ and $\hat{B}$ then appear as parameters of our main optimization in (26) to (29), which are constraints to ensure that the LQR solution is equal to the target policy (our attack goal). Constraint (31) is simple. We point out that when converting the follower problem (30) into its equivalent KKT condition, one should incorporate its decision variables into the leader-level optimization since there will be no follower problem anymore, which is why we write decision variables for both leader and follower problem on the LHS of (25). We will make it more clear in the revised version.

*(2) scalability issue.* We agree that the scalability issue is a weakness of our paper. However, since we are aiming for \*global optimality\* in our attack, we require the RL learner to take the form of some convex optimization (e.g., LP in discrete MDP case), which can be replaced with its KKT equivalent. For more complex learners such as deep RL learners, as they rely on highly non-convex neural networks, the KKT condition approach may not apply. In that case, solving the global optimality using convex optimization might not be feasible, but approximate methods such as gradient descent can be used to find locally optimal solutions. We agree scalability is important, and we will discuss it in the revised paper. We leave that as future work, as our main goals of the paper is to first formally define the problem of adversarial poisoning in RL/control and study if \*globally optimal attack\* is achievable.

We actually also experimented on a continuous domain: Linear Quadratic Regulator, which is a classic continuous control problem and arguably represents many real world control problems.

# References

[1] Shike Mei and Xiaojin Zhu. Using machine teaching to identify optimal training-set attacks on machine learners. In *Twenty-Ninth AAAI Conference on Artificial Intelligence*, 2015.


[Meta-Review · NeurIPS 2019]

Dear authors: your paper was carefully evaluated by the reviewers, and was discussed after we received the rebuttal. There was general agreement that this was an interesting paper and worthy of acceptance at NeurIPS 2019. Adversarial attacks on policy learning in RL is very timely. I would like to note, however, that I solicited some outside feedback on this paper after the reviews were in, and this feedback had both positive and negative comments. This 4th perspective was, I think, particularly on point and worth reading carefully, and I will share it below. I would like to encourage the authors to take this, and the other reviews, into account when preparing their final submission. ==== Additional Feedback to Authors ==== - This paper looks like an extension of the previous work on data poisoning attacks [15] (from bandit to RL) in the sense that it uses the same problem formulation (reward modification towards a target policy and l_p norm as a "cost"). - Although this is a novel extension with a theoretical contribution, the theoretical/empirical result is quite limited to offline batch RL and simple algorithms (tabular / LQR), which is far from real applications. I think the RL community is generally more interested in online learning, where the agent keeps collecting data, and more complex RL architectures/algorithms (e.g., policy gradient with linear function approximation). At least, it could be better if this paper justified how realistic this scenario is. - (Minor comment) Introducing the term "policy poisoning" is misleading because the problem setup is exactly the same as "data poisoning" paper [15], which only manipulates data (not the policy itself). Also, the paper should refer to the previous work when describing the problem setup and objective and clarify this in the related work.